# Multifunctional Photoactive Nanomaterials for Photodynamic Therapy against Tumor: Recent Advancements and Perspectives

**DOI:** 10.3390/pharmaceutics15010109

**Published:** 2022-12-28

**Authors:** Rupesh Jain, Shambo Mohanty, Ila Sarode, Swati Biswas, Gautam Singhvi, Sunil Kumar Dubey

**Affiliations:** 1Department of Pharmacy, Birla Institute of Technology and Science-Pilani, Pilani Campus, Pilani 333031, Rajasthan, India; 2Department of Pharmacy, Birla Institute of Technology and Science-Pilani, Hyderabad Campus, Hyderabad 500078, Telangana, India

**Keywords:** photodynamic therapy, cancer, nanocarriers, actively targeted drug delivery

## Abstract

Numerous treatments are available for cancer, including chemotherapy, immunotherapy, radiation therapy, hormone therapy, biomarker testing, surgery, photodynamic therapy, etc. Photodynamic therapy (PDT) is an effective, non-invasive, novel, and clinically approved strategy to treat cancer. In PDT, three main agents are utilized, i.e., photosensitizer (PS) drug, oxygen, and light. At first, the photosensitizer is injected into blood circulation or applied topically, where it quickly becomes absorbed or accumulated at the tumor site passively or actively. Afterward, the tumor is irradiated with light which leads to the activation of the photosensitizing molecule. PS produces the reactive oxygen species (ROS), resulting in the death of the tumor cell. However, the effectiveness of PDT for tumor destruction is mainly dependent on the cellular uptake and water solubility of photosensitizer molecules. Therefore, the delivery of photosensitizer molecules to the tumor cell is essential in PDT against cancer. The non-specific distribution of photosensitizer results in unwanted side effects and unsuccessful therapeutic outcomes. Therefore, to improve PDT clinical outcomes, the current research is mostly focused on developing actively targeted photosensitizer molecules, which provide a high cellular uptake and high absorption capacity to the tumor site by overcoming the problem associated with conventional PDT. Therefore, this review aims to provide current knowledge on various types of actively and passively targeted organic and inorganic nanocarriers for different cancers.

## 1. Introduction

Cancer is the second biggest cause of mortality and, therefore, is a serious public health concern worldwide [1]. The early therapeutics and traditional therapies mostly aim to reduce the incidence rates of cancers, but the basic problem with curing cancer patients with conventional therapies is that these treatments have a low selectivity for cancer cells and hence must be given at high lethal chemical doses to be efficacious [2]. When patients receive conventional types of treatment, e.g., chemotherapy or radiation, the high amount of toxic drugs is likely to harm normal body cells as well, causing serious side effects. Furthermore, tumor heterogeneity, drug resistance, and systemic toxicity cause significant barriers to most conventional cancer treatments [3,4].

Several management techniques are available to treat cancer, such as chemotherapy, radiation immunotherapy, surgery, stem cell or bone marrow transplant, and hormone therapy [5]. Photodynamic therapy (PDT) is a comparatively recent treatment strategy that has been examined in greater depth. PDT is a clinically approved, non-invasive novel therapy used to treat various cancers [6,7,8,9]. PDT treatment involves a group of essential molecules called photosensitizers (PS). When PS molecules are irradiated at a specific wavelength of light, a tremendous amount of energy is transferred to oxygen molecules or other substrates in the surrounding area of the tumor to form highly reactive oxygen species (ROS), such as singlet oxygen (^1^O_2_), resulting in cell death by the mechanism of cellular autophagy, cellular apoptotic and cellular necrosis [10,11,12,13]. At first, the photosensitizer is injected into blood circulation or applied topically, where it quickly becomes absorbed or enters the cells. The delivery of nanoparticles containing PS is based on the difference between the physiology of tumor cells and normal cells. Due to the enhanced permeability of tumor blood vessels and disrupted lymphatic drainage, nanoparticles reside in tumor cells [14,15,16]. Then, the tumor is irradiated with a specific wavelength of light which leads to activating the PS molecule and generating the ROS, resulting in the death of tumor cells and therapeutic effects in PDT [12,17,18]. The application of PDT on a solid tumor is schematically represented in Figure 1.

There are three potential ways a tumor can be destroyed by PDT. In the first mechanism, the tumor can be destroyed during light irradiation to the tumor, and ROS species will become engendered and kill the cancer cells by the mechanism of apoptosis, autophagy, and necrosis. In the second mechanism, PDT destroys tumor cells by attacking the vasculature and tumor environment. In the third mechanism, tumors can be destroyed by the generation or activation of an immune response. A few factors involved in the activation of the immune response are inflammatory mediators or T- helper cells (Figure 2) [19,20,21,22]. Additionally, significant pathways of proteolytic cleavage of cellular substrates by effector caspases include one pathway where the mitochondria release protein into the cytoplasm to activate caspase 9. Caspase 9 activations inside the mitochondria may have been triggered by stressful mitochondrial environmental conditions. Apoptogenic proteins such as cytochrome c are released in the cytosol of the mitochondria from the intermembranous space under stress. Cytochrome c further makes a complex with apoptotic protease activating factor 1 (Apaf-1), procaspase 9, and dATP/ATP (apoptosome) by acting as a co-factor. This complex activates caspase 9, which comes out of the mitochondria into the cytoplasm and triggers cleavage of executioner caspases such as caspase 3, 6, and 7. These caspases inactivate the proteins that work to prevent cellular apoptosis. Such inactivation inevitably leads to mitochondria-mediated apoptosis of tumor cells in PDT. There are other caspases, e.g., caspase 8, however, with PDT, only caspase 9 activity has been observed predominantly [23,24].

There are several pathways to reach such a fate. More precisely, PDT triggers a wide range of changes in cellular metabolism that ultimately leads to cell death and tumor destruction. Either the cells follow a caspase or cytochrome-c mediated mitochondrial pathway or follow a pathway involving ceramides or death receptors.

PDT has been shown to increase the cellular Ca2+ levels via ion channel influx, activation of ion exchange mechanism, etc. High Ca2+ levels in the cytoplasm activate phospholipase A2. This pathway is of great significance in cellular apoptosis [23]. Furthermore, the mitogen-activated protein kinase (MAPK) pathway modulates the extracellular signal-regulated kinase pathway (ERK-1/2). Upon PDT, a steep decrease in ERK expression was observed, which was directly related to the sharp increase in cell death. Transcriptor factors may also have some effects in dictating the cell death pathway by PDT. The activation of promoter regions in genes by chemical as well as physical stresses induced by PDT has been shown to be controlling apoptosis induction in cells. Several PDT studies have indicated the involvement of PDT in the expression of cytokines inside the cells. The overexpression or underexpression of these proteins dictates tumor regression as well as tumor eradication via changing the membrane-protein expression and the proliferation of effector molecules. Another possible PDT-induced cell death pathway includes heat shock proteins (Hsps) and glucose-regulated proteins (Grps). A few studies have postulated that the light activation of photosensitizers triggers a specific gene expression that is directly associated with Hsps and Grps. This further decides if the cell will die or develop resistance against PDT. Another gene expression that may have a significant role in cellular apoptosis is the hypoxia-inducible factor (HIF-1). As PDT induces hypoxia rapidly in the tumor microenvironment by cutting off the blood supply and using excess oxygen in the cells to activate the photosensitizer molecules, an association of HIF-1 in PDT is probable However, further investigation is required to establish this pathway as a legitimate PDT-triggered cell death pathway in cancer therapy [23,25,26].

In comparison to conventional therapy, PDT has a lower occurrence of side effects, and it also increases the target specificity. Although PDT has many advantages over standard cancer therapies, it still has yet to be wildly utilized in clinical health care. PS molecules possess several intrinsic properties such as poor targeting, low solubility, and high hydrophobicity because of which they easily form aggregates in physiological conditions [27,28,29]. Even though several PS molecules have been modified to increase water solubility, their selectivity of the target site is still inadequate, leading to their unsuccessful clinical uses. Table 1 represents the clinical trials and approval status of various photosensitizers [30,31,32,33,34,35,36,37,38,39,40,41,42,43,44].

In this context, the fabrication of an effective strategy is required for delivering PS molecules at the specific site or targeted area of the tumor, while overcoming the critical obstacle associated with conventional therapies. The combination of actively targeted nanomaterials with PDT therapy has shown extensive interest in treating cancer with high selectivity and specificity. The above actively-targeted functionalized nanomaterials enhance the rate of treatment outcomes, patient compliance and reduce the problems associated with conventional therapy or traditional therapeutics [45,46,47,48,49].

Recently, nanomaterials have shown significant promise in order to maximize PS solubility and targeting, which enhances the therapeutic outcomes of PDT. PS molecules can easily achieve passive targeting with the help of enhanced permeation and retention (EPR) effect when fabricated as nanoparticles. Further, surface modification of nanoparticles by binding with active targeting moieties such as functionalized groups, antibodies, peptides, and aptamers can extensively improve the cell target specificity of PS molecules [50,51,52,53,54].

Numerous nano-platforms based on a wide range of organic and inorganic nanomaterials have been developed over the year for effective uptake and targeted PS delivery [27,55]. Among the organic nanomaterials liposomes, polymeric nanoparticles, solid lipid nanoparticles, nanostructured lipid carriers, etc., have been evaluated for delivering PS molecules to the site of action and have been found effective, and safe delivery has been achieved [56,57]. The use of organic nanocarriers has successfully achieved flexibility in the formulation, such as increasing bioavailability, permeability, tumor accumulation, etc. Inorganic nanoparticles also have significant advantages over organic nanoparticles because of their high stability, ease of surface modification to make them suitable for specific targeting (specific ligand), tunable size, and optical properties. They are also more useful in diagnosis or imagining purposes [58,59].

Different types of organic and inorganic nanoparticles have shown significant effects. They both provide massive application in drug therapeutics by overcoming the problem associated with the PS molecules, such as instability at physiological conditions, improper specificity, high hydrophobicity, etc. In addition, both organic and inorganic nanoparticles are explored in combination with PS via chemical modification or chemical bonding, electrostatic interaction, and conjugation process for cancer treatment.

## 2. Targeting Photosensitizers via Active/Passive Targeting

Cellular uptake of PS molecules from subcellular localization can be categorized as either passive or active targeting. Passive targeting mainly occurs via the EPR effect of tumor-associated leaky vasculature and lymphatic drainage of tumor tissues. Moreover, target cell specificity of PS molecules has been proved to be efficient when those are attached to the surface-active moieties including aptamer, antibodies, peptides, small ligands, folic acid, etc. It was also observed to increase the therapeutic effect and cytotoxicity against the cancer cells. Active targeting has significant advantages over passive targeting because of its more selective target specificity, and tendency to accumulate more PS molecules in the intracellular tumor environment [46,60,61,62].

Passive targeting is based on the difference between the physiology of tumor cells and normal cells. Due to the enhanced permeability of tumor blood vessels and disrupted lymphatic drainage, nanoparticles reside in tumor cells. Passive targeting is not found to be effective in all cases, e.g., early-stage tumors do not show an EPR effect. Efficacy and specificity are compromised in passive targeting as the permeability of blood vessels is not the same all over the tumor cells. Active targeting has the potential to overcome the disadvantages of passive targeting. Targeting ligands are attached to the photosensitizer or the nanocarriers containing the photosensitizer, which binds to the receptors or antigens that are overexpressed on the tumor cells. Two types of active targeting are tumor targeting and vascular targeting. Tumor targeting refers to the ligands binding to specific receptors, while vascular targeting refers to the ligands binding to the receptors present on the endothelial cell surface. The uptake of nanoparticles by tumor cells is enhanced due to tumor targeting. In contrast, in vascular targeting, the concentration of nanoparticles is enhanced in vascular tissue, even in non-leaky vessels. Vascular targeting helps to kill the tumor by targeting blood vessels responsible for the supply of oxygen and nutrients. The tumor cell markers include folic acid receptors, integrin receptors, transferrin receptors, low-density lipoprotein receptors, glucose transporters, and growth factor receptors [63,64]. Antibody-based targeting has been considered one of the best strategies for drug delivery due to its antigen specificity. Photoimmunotherapy refers to the tumor-targeting strategy in which the antibody fragment or antibody is coupled to the photosensitizer for targeting the tumor antigen. Hydrophilic photosensitizers are preferred for photoimmunotherapy [63,65]. Aptamer-based targeting has been explored for targeting cancer cells in photodynamic therapy. Aptamers are three-dimensional structures of short nucleotides that can bind to membrane receptors and extracellular and intracellular targets. They are found to bind with high specificity and affinity to protein as well as non-protein targets. They exhibit unique features, including lack of immunogenicity, structural stability, ease of isolation, and small size. The single-stranded deoxyribonucleic acid (DNA) or ribonucleic acid (RNA) molecules can regulate the function of genes by binding to the target sites and beginning replication and translation [63,65,66]. Peptides are amino acids linked by the amide bond. The amino acids present in the peptide are less than fifty, and hence the peptide has a low molecular weight. This property leads to an indefinite tertiary structure and good receptor binding affinity. However, they are subjected to peptidases and rapid clearance. Several approaches have been introduced to overcome this problem, including developing modified peptides that are less susceptible to peptidases, thereby enhancing their bioavailability. A modification of the amino acids can be performed, which enhances their in vivo stability. They have several advantages, such as they form non-toxic degradation products since they are made of natural amino acids. These peptides can alter their confirmation to secondary structure in response to changes in pH or temperature. Folic acid is one of the many vitamins which has promised its use in targeted photodynamic therapy. Its stability, non-toxicity, non-immunogenicity, and low cost have further enhanced its use in PDT. It can be coupled with photosensitizers quite easily. It has an affinity for folate receptors which are overexpressed in several cancers, including lung, brain, kidney, breast, and ovary [63,65,67].

## 3. Inorganic and Organic Nanoparticulate for Active or Passive Photosensitizer Drug Delivery

### 3.1. Inorganic Nanoparticulate for Active or Passive Photosensitizer Drug Delivery

In the last decade, the nanotechnology sector has grown and has been explored at a rapid rate, resulting in the development of potential nanocarriers for applications in various areas such as medicine, electronics, biomaterial engineering, etc. Nanomedicine is the use of nanotechnology in biomedical research, and it involves the use of nanoparticulate that have been carefully designed for novel diagnostic and drug delivery purposes to enhance therapeutic efficacy [68,69,70].

In a nanoparticulate drug delivery system, PS are encapsulated with polymer or lipids or a ligand-based system with covalent bonding or noncovalent interaction. A few significant advantages of attaching the nanoparticles with the PS molecules are the high drug loading capacity, and a high surface-to-volume ratio may be observed. The nanoparticulate system is attractive in PDT therapy for the following points: first, PS molecule concentration becomes enhanced at the desired targeted site because of targeting potential and reduces the unwanted toxic effect on healthy cells. Second, the solubility of hydrophobic PS molecules can be significantly improved by the nanoparticulate system. Third, the nanoparticulate system can deliver the drug at the desired targeted site in a sustained and controlled manner. Therefore, the functionalization of the nanoparticle platform is used to enhance the effectiveness of PDT and to deliver the PS molecules at the desired targeted site by utilizing the active or passive mechanism. PDT with actively targeted nanoparticles consists of numerous types of organic and inorganic nanoparticles, including liposomes, lipid-based nanoparticulate, polymeric micelles, carbon dots, gold nanoparticles, magnetic nanoparticles, quantum dots, etc. [71,72,73,74,75]. Some recent major outcomes of nanoparticulate in the delivery of PS to the desired site of the tumor by utilizing the overexpressed receptors site and their ligands are mentioned in Table 2 [76,77,78,79,80,81,82,83,84,85,86,87,88].

Inorganic nanoparticles are preferred over organic nanoparticles as they show several advantages, including tunable size, high stability, ease of surface modification, and optical properties. They exhibit lower degradation as compared to the organic nanoparticles [89]. They include quantum dots, gold nanoparticles, mesoporous silica nanoparticles, graphene, fullerene, and magnetic nanoparticles. The PS in these non-biodegradable nanostructures produces singlet oxygen without them being released. However, in organic nanoparticles, the singlet oxygen species must have the ability to diffuse across these nanostructures to produce the desired photodynamic effect. [63]. The intrinsic functions of inorganic nanoparticles and the special characteristics of each of these nanoparticles have attracted the attention of the scientific community in biomedical applications. Gold nanoparticles show a photothermal effect due to surface plasmon resonance; iron oxide nanoparticles can be utilized as T2 magnetic resonance imaging contrast agents, and mesoporous silica nanoparticles have high loading capacity, large surface area, and facile size control [49,90,91].

#### 3.1.1. Silica Nanoparticles

Mesoporous silica nanoparticles are utilized in photodynamic therapy because of their pore volume, large surface area, and biocompatible nature. Photosensitizers can be covalently linked to the silica nanoparticles, or they can be encapsulated. These nanoparticles facilitate the attachment of ligand and stimuli-responsive substances that can be integrated to release the drug at the diseased site [92]. Ozge et al. developed Cetuximab-targeted mesoporous silica nanoparticles capped with imidazole for the delivery of zinc phthalocyanine to treat human pancreatic cancer. The cell viability of targeted mesoporous silica nanoparticles was found to be 6.2%, 12.5%, and 17.5% for PANC-1, AsPC-1, and MIA PaCa-2 cells, respectively while that of zinc(II) 2,3,9,10,16,17,23,24-octa(tert-butylphenoxy))phthalocyaninato(2-)-N29,N30,N31,N32 (ZnPcOBP) was found to be 35%, 55%, and 39%. The increase in efficacy was attributed to cetuximab’s targeting effect, which recognizes the cells and internalizes the cells through receptor mediation [93]. Planas et al. synthesized amino and mannose-targeted mesoporous silica nanoparticles loaded with methylene blue (MMSNP-MB) and evaluated their antibacterial activity. MMSNP-MB decreased 8-log10 the bacterial survival fraction on exposure to 16 J/cm^2^ while amino- or mannose-modified mesoporous silica nanoparticles (AMSNP-MB) showed a reduction of 5-log10 on exposure to 32 J/cm^2^. It was concluded that the photodynamic effect shown by MMSNP-MB and AMSNP-MB was similar. However, for E.coli, dark toxicity was improved, and mannose proved to be a better target for P. aeruginosa than amino groups [94]. Ma et al. developed folic acid functionalized hollow mesoporous silica nanoparticles for photodynamic therapy containing aminolevulinic acid to treat skin cancer. Results indicated accumulation of protoporphyrin IX (PpIX) in skin cancer cells through folate receptor-mediated endocytosis. These nanoparticles were activated at 635 nm and released Aminolevulinic acid (ALA), which killed skin cancer cells because of the conversion of ALA to PpIX [95].

#### 3.1.2. Gold Nanoparticles

Gold nanoparticles are employed in medicine because of their photophysical, optical, and photochemical properties. In addition, gold nanoparticles are biocompatible and show low toxicity. They are employed in photodynamic therapy as they offer several benefits, including an increase in singlet oxygen production, their potential to become conjugated to biological ligands, and the dispersion of hydrophobic photosensitizers in aqueous media. Several types of gold nanoparticles can be employed in photodynamic therapy, including spherical-shaped nanoparticles, nanoclusters, nanorods, and nanostars. Gold nanorods offer the advantage of combining photodynamic therapy with photothermal therapy. Photosensitizers are coupled to gold nanoparticles with a different approach which is based on self-assembly. Electrostatic interactions and covalent binding are some of the techniques for preparing these metallic nanoparticles [96]. Goddard et al. developed peptide-directed gold nanoparticles containing phthalocyanine for photodynamic therapy. The IC50 of zinc phthalocyanine C11Pc poly(ethylene glycol) (PEG) gold nanoparticles (peptide-C11Pc-PEG-AuNps) was found to be 105.8 nM on irradiation while >250 nM without irradiation. No phototoxic response was recorded when the cells were incubated with non-targeted C11Pc-PEG-AuNps. It was concluded that the phototoxicity of peptide-C11Pc-PEG-AuNps was due to the dual effect of photocatalytic activity of C11Pc and the targeting effect [97]. Shiao et al. developed gold nanoparticles modified with aptamer for co-drug delivery in photodynamic therapy. It was found that the therapeutic efficacy of 5,10,15,20-tetrakis(1-methylpyridinium-4-yl) porphyrin (TMPyP4)-loaded dox-NPs was raised to 4.6 and 2.5-fold as compared to chemotherapy and photodynamic therapy alone. This depicted that a combination of the treatment majorly improved the antitumor efficacy. Nanoparticular delivery ensures effective intracellular transport and overcomes tumor drug resistance. Thereby, it increases antitumor effectiveness [98].

#### 3.1.3. Carbon Nanomaterials

Carbon nanomaterials are of different types based on their dimensions, such as fullerenes are zero-dimensional, carbon nanotubes are one-dimensional, and graphene nano molecules are two-dimensional. They exhibit unique physicochemical properties and structures that do not cause chemotherapeutic toxicity and are effective in novel therapies. Carbon nanotubes and graphene are generally used in delivery systems due to their large surface area and loading capacity of PS and drugs [99]. They show high mechanical strength, good optical properties, biocompatibility, and low toxicity, which has promoted their use in the biomedical field. They are used in photodynamic therapy due to their unique properties, such as C60 acting as a photosensitizer. Carbon nanotubes can convert the near infra-red light into heat. They display high drug loading because of pi-pi stacking. However, there is a need to reduce their dose-dependent toxicity and increase aqueous solubility [100].

#### 3.1.4. Carbon Nanotubes

The distinct chemical and physical properties of carbon nanotubes have attracted their attention in biomedical applications. PS can be conjugated to carbon nanotubes to enhance the solubility, targeting, and bioavailability of PS [101]. They have the ability to transform light into heat which can be used to design thermal therapy. They can load hydrophobic drugs and absorb light strongly in the near IR region, enhancing the penetration of light to deeper tissues. As compared to iron oxide nanoparticles, they show a high heating rate and there is no need to remove metallic material [102]. Zhang et al. developed single-walled carbon nanotubes coated with aptamer conjugated magnetofluorescent iron oxide carbon quantum dots for chemo/photodynamic/photothermal triple modal therapeutic agents for the treatment of lung cancer. The aptamer targeting enhanced cellular uptake. The aptamer conjugated PEG 2000 N modified Fe3O4@carbon quantum dot coated single-walled carbon nanotubes (SWCNTs-PEG-Fe_3_O_4_@CQDs/doxorubicin (DOX) exhibited more cytotoxicity against HeLa cells as compared to that of non-targeted. It was concluded that the developed SWCNTs showed multiple effects such as NIR photothermal heater, drug carrier, combination therapy of cancer, and multimodal imaging probe [103]. Shi et al. developed hyaluronic acid-modified carbon nanotubes (HA-CNT) containing hematoporphyrin monomethyl ether (HMME) for photodynamic therapy. The rate of inhibition of HMME-HA-CNTs at a concentration of 12.5 µg/mL was found to be 76.8% with 532/808 nm laser and 41.1% for HA-CNT with 808 nm laser. It was found that the photodynamic and photothermal effect of HMME-HA-CNTs was more than that of photodynamic/photothermal therapy given alone [104].

#### 3.1.5. Fullerene

Fullerenes are employed in photodynamic therapy because of their photostability and undergo relatively less photobleaching than tetrapyrroles. It is easy to modify fullerenes and achieve the desired lipophilicity. Light-harvesting antennae are conjugated to fullerenes to enhance the generation of reactive oxygen species. Fullerenes can undergo self-assembly and form fullerosome. This fullerosome possesses targeting properties and can function as multivalent drug delivery vehicle. The core of fullerene is composed of C60 to which amphiphilic or fused ring structures or hydrophilic side chains can be attached. However, fullerenes absorb in the ultraviolet or blue, or green regions of the spectrum, which limits the penetration of light to deeper tissues. Some approaches have been developed to overcome these limitations, such as the conjugation of red light-absorbing antennae to the core. Optical clearing agents can also be utilized to overcome this limitation [105]. The core of fullerene, composed of C60, undergoes a transition from the S0 state to short-lived the S1 state. The excited state decays at 5 × 108 sec-1 and undergoes intersystem crossing to the T1 triplet state, which has a long lifetime. The triplet state of fullerene is quenched due to the presence of molecular oxygen, which is present as a triplet in the ground state. The quenched triplet state of oxygen generates singlet oxygen by energy transfer. This process generates a singlet oxygen quantum yield near the theoretical maximum of 1 [105,106]. Liu et al. developed R13 aptamer conjugated trimalonic acid modified C70 fullerene to increase the effectiveness of photodynamic therapy for the treatment of lung cancer. The cell viability shown by TF70-R13 was 0.1 while that of TF70 and control was 0.4 and 1, respectively [107]. Zhang et al. formulated supramolecular hydrogel of dipeptide fullerene for antibacterial photodynamic therapy. Hydrogel avoids the aggregation of fullerene because of the non-covalent interactions between fullerene and peptides. It was observed that 42% of bacteria were killed by C-60 pyrrolidine tris acid while no bacteria were found on treatment with peptide fullerene and light irradiation. It was concluded that the non-covalent interactions have a synergistic effect on hydrogel properties. The peptide nanofibers enhance the singlet oxygen generation capacity. The fullerene nanoparticles are responsible for enhancing the mechanical properties of the hydrogel, thereby ensuring a targeted effect [108]. Shi et al. prepared doxorubicin (DOX) fullerene nanoaggregates nanoparticles and attached the hydrophilic moiety, distearoyl-sn-glycero-3-phosphoethanolamine-PEG-CNGRCK2HK3HK11 (DSPE-PEG-NGR) (C60-DOX-NGR-NP) which showed high antitumor efficacy and low toxicity to the normal cells as shown in fig 4T1 cell targeting ability of C60-DOX-NGR-NP, C60-DOX-PEG NP, and DOX was evaluated and results indicated higher red fluorescence in case of C60-DOX-NGR-NP as compared to C60-DOX-PEG-NP, suggesting its higher tumor targeting ability. In vitro studies revealed the highest decrease in cell viability in the case of C60-DOX-NGR-NP to 58.7% on the application of laser light. The in vivo studies indicated the highest concentration of doxorubicin in tumors from C60-DOX-NGR-NPs and inhibited tumor growth after applying the laser light twice [109].

#### 3.1.6. Graphene

Scientists have used graphene and graphene oxide to explore their role in the treatment of cancers and photodynamic therapy. Graphene nanoparticles are classified as graphene oxide, single-layered graphene, reduced graphene oxide, and few-layered graphene. Graphene oxide (GO) has a large specific surface area that ensures the effective loading of photosensitizers and drugs with the help of surface functional groups. Graphene nanoparticles in PDT have been proved to be more stable, bioavailable, and photodynamically efficient. Its unique optical, mechanical, and electronic properties have enhanced its use in biomedical science [110,111]. Graphene oxide increases the photodynamic activity of inorganic nanoparticles such as Titanium dioxide (TiO_2)_ and Zinc oxide (ZnO). These inorganic nanoparticles produce reactive oxygen species like hydroxyl radicals, superoxide radicals, and hydrogen peroxide by irradiating them with UV light. UV light has a major limitation in that it cannot penetrate deeper tissues. The ROS which is generated by this type of light has a short life span, which lowers the antitumor efficacy. Hu et al. reported that graphene oxide on the surface of TiO_2_ could activate visible light responsive activity. Graphene exhibits high electrical conductivity, which transfers the light-triggered electron by TiO_2_ to graphene oxide. This process suppresses electron-hole recombination. The holes in the valence band of TiO_2_ move to the surface and interacts with water to form hydroxyl radicals. The electron transferred by TiO_2_ to GO reacts with oxygen to form singlet oxygen. In this way, GO can increase the photodynamic efficiency of inorganic nanomaterials [111]. Wei et al. developed the integrin α_v_β_3_ functionalized nanosystem of graphene oxide (NGO). The surface of the NGO was covered with polyethylene glycol, which was conjugated to pyropheophorbide. Pyropheophorbide-a (PPa) provided a phototoxic effect. The PPa only PDT treatment showed a cell death of 50%, while α_v_β_3_ NGO resulted in 70% of cell death. The results were attributed to the targeting activity of the antibody and enhanced drug loading/transporting ability. Huang et al. formulated folic acid conjugated graphene oxide containing the photosensitizer chlorin e6. Loading of chlorin e6 in GO was carried out by pi-pi stacking and hydrophobic interactions. Cytotoxicity results indicated that FolicAcid-GO was non-toxic to cancer cells. A 1:1 ratio of mFA-GO/m-Ce6 showed a cell viability of less than 50%. It was concluded that the developed novel nanosystem could be formulated with low cytotoxicity and improved solubility. The developed system increased the amount of photosensitizer in the cancer cells [112].

#### 3.1.7. Iron Oxide Nanoparticles

Iron oxide nanoparticles have been utilized in biomedicine due to their targeting properties, biocompatibility, superparamagnetic nature, easy surface modification, and small size. Iron oxide nanoparticles were developed for delivering a combination of photodynamic therapy and magnetic resonance imaging [113]. Zeng et al. developed folic acid-targeted superparamagnetic iron oxide nanoparticles containing photosensitizers (FA-NPs-PS) for visualized photodynamic therapy and magnetic resonance imaging. Fluorescence of photosensitizer was noted in MCF-7, HeLa cells, and MCF-7 tumors proving that folic acid showed a good targeting effect. The results indicated that the cell viability in MCF-7 and HeLa cells treated with FA-NPs-PS was 18.4% and 30.7%, while that of the cells treated with NPs-PS was 25.7% and 34.4%, respectively. MCF-7 tumors showed an inhibition of about 94.9%. It was concluded that the prepared nanoparticles were effective magnetic resonance imaging (MRI)/PDT nanoprobes for visualized therapy of breast cancers and in vivo diagnosis [114]. Patel et al. developed novel folic acid-modified iron oxide-zinc oxide nanoparticles as a photosensitizer in photodynamic therapy. Zinc oxide is an n-type semiconductor that exhibits high photosensitivity, low cost, and is environmentally friendly. Zinc oxide undergoes photoexcitation when subjected to UV light which has higher energy than its bandgap energy. This photoexcitation results in the formation of positive holes and negative electrons in the valence band and conduction band, respectively. The electron-hole pair possesses reduction and oxidation properties that recombine or are captured by water or oxygen, leading to ROS formation. The hybrid iron oxide-zinc oxide nanoparticles enhanced photophysical properties due to the decrease in charge recombination in zinc oxide by the presence of iron oxide, which acts as electron trapping sites. The photokilling effect was enhanced by conjugating folic acid, which reduced cell viability to 34% on irradiation with FZ-SFA50. It was concluded that the synthesized nanoparticles showed effectiveness in photodynamic therapy [115].

#### 3.1.8. Quantum Dots

They are used in multifunctional nanocarriers for photodynamic therapy because of their facile surface modification, high emission quantum yield, and tunable optical properties. The size of the quantum dots from 1 to 6 nm offers them antique optical properties that can be changed from ultraviolet to infrared by adjusting the composition and size. The quantum confinement effect regulates the emission properties, and they can be tuned to emit in the infrared region. This is in opposition to that the visible emission of most photosensitizers. This effect allows deep penetration of light and can be used to cure deep tumors. This deep penetration of light is attributed to the minimal scattering and absorption of light in the near-infrared region. Quantum dots (QD) exhibit large transition dipole moments, making them strong absorbers and facilitating their use in photodynamic therapy [116]. Li et al. formulated peptide-targeted quantum dots to be used in photodynamic therapy to treat pancreatic cancer. The relative tumor volume after treatment with RGD-QD was found to be 3.24, while that of PDT treatment without injection of RGD-QD was found to be 7.25. It was concluded that RGD-QD in photodynamic therapy was found to be successful in the treatment of pancreatic cancer [117]. Cao et al. developed aptamer conjugated graphene quantum dots (GQD) for photodynamic and photothermal synergistic therapy and as the diagnostic agent of cancer-related micro-RNA. The GQD-PEGP treated cells showed cell viability of 14%, while the control groups recorded a cell viability of 90%. It was concluded that a novel theranostic agent of aptamer-conjugated PEGylated GQD loaded with porphyrin derivatives was successfully developed [118].

### 3.2. Organic Nanoparticulate for Photosensitizer Drug Delivery

#### 3.2.1. Liposomes

*Liposomes* are used to enhance the concentration of photosensitizers at the tumor site in photodynamic therapy due to their high loading capacity. They can also encapsulate photosensitizers of different physicochemical properties. Actively targeted liposomes show enhanced plasma half-life, thereby enhancing efficacy [119]. Active targeting of liposomes ensures increased photosensitizer accumulation at the tumor site and increases the photodynamic effect. It eliminates the undesired side effects of the photosensitizer. The binding site barrier is the limitation of targeted liposomes. The targeted liposomes bind to the target that they first encounter, thus retarding their penetration. However, PDT is used for the treatment of superficial tumors because of the limited penetration of light. Hence, not much attention needs to be paid to this phenomenon [56]. Peptide-targeted liposomes exhibit less immunogenicity and small molecular weight as compared to antibodies. They can be synthesized, stabilized, and modified easily. However, they have less binding affinity as compared to antibodies [120]. Huang et al. developed GE11 Peptide conjugated liposomes containing indocyanine green and curcumin for epidermal growth receptors targeting cancer cells. It was reported that the cell viability was reduced by curcumin/indocyanine green and curcumin/indocyanine green-liposomes was 52% and 35.2%, respectively while GE11-curcumin/indocyanine green-liposomes reduced cell viability to 12%. It indicated a greater cytotoxic and phototoxic effect on cancer cells [121]. Antibody-targeted liposomes identify the antigens which are overexpressed on the tumor cells and exert a targeting effect. They are the most widely used targeting agents. [120]. Broekgarden et al. developed an approach for site-specific conjugation of EGa1 sdAbs antibodies on liposomes encapsulating zinc phthalocyanine targeting the epidermal growth factor receptor. It was found that the antibody-conjugated liposomes exhibited selective uptake characteristics and enhanced the photodynamic efficacy in comparison to the non-targeted liposomes [122]. In a study, Anilkumar et al. prepared dual-targeted dual-mode nano-vehicles. Targeting properties were induced by magnetic, as well as, ligand-based approaches while dual-mode was achieved via photothermal and photodynamic therapies. 1,2-distearoylsn-glycero-3-phosphocholine, dimethyldioctadecyl ammonium bromide (DDAB), and cholesterol were used to prepare magnetic photosensitive liposomes (MPLs) that encapsulated the PS molecules (indocyanine green), and Fe_3_O_4_ (coated magnetic nanoparticles or CMNPs). Then, the MPLs were coated using Hyaluronic acid-polyethylene glycol (HA-PEG) to ultimately form HA-PEG-MPLs. The coating was performed by utilizing the negative charge of HA and the innate positive charge of DDAB. Self-assembly of HA-PEG over MPLs was observed. The resulting NPs showed promising PTT and PDT effects when irradiated by successive NIR laser regimens. A nude mice model was used to show the effects where this novel strategy resulted in ~7.8-fold reduction in tumor volume compared to the control at the end of the treatment [123]. Wohrle et al. prepared Zn(II)-2,3-naphthalocyanines including four zinc naphthalocyanines (ZnNc), the unsubstituted ZnNc 1, tetraacetylamido substituted ZnNc 2, tetraamino-substituted ZnNc 3, and tetramethoxy substituted ZnNc 4 and loaded them in liposomes for photodynamic therapy. The pharmacokinetic properties of ZnNc 1 dipalmitoylphosphatidylcholine liposomes and the photodynamic effect of ZnNc 2-4 were studied by their intraperitoneal administration in hamsters infected with rhabdomyosarcoma. The phototherapeutic effect was determined by mean tumor diameter, photonecrosis, and the percentage of animals recovered. ZnNc 2 showed the highest percentage (50%) of animals recovered. ZnNc 4 recovered 40% of animals while ZnNc 3 did not show any effect upon photodynamic therapy and studies on ZnNc 1 indicated a very low effect. Zn(II)-2.3-naphthalocyanines were found to be effective at low doses and show selective targeting and slow clearance from the tumor [124].

Reddi et al. studied the pharmacokinetic properties and phototherapeutic effect of Zn (II)-phthalocyanine (ZnPc) loaded in dipalmitoylphosphatidylcholine and low-density lipoprotein liposomes by injecting them intravenously in mice infected with MS-2 fibrosarcoma. The photodynamic effect at various doses of irradiation was measured by determining the efficiency of tumor necrosis which indicated that 0.07–0.35 mg/kg dose of Zn-Pc is enough for producing an effective tumor response. On increasing the irradiation from 50 to 200 mW/cm^2^, a six-fold increase in necrosis was observed. Zn-Pc was found to be an effective photodynamic agent because of its slow clearance, local effect, and high tumor necrosis efficiency at low doses [125].

#### 3.2.2. Solid Lipid Nanoparticles

Solid lipid nanoparticles are used in photodynamic therapy due to their biocompatibility, less toxicity, and protection of photosensitizers from the external environment [126]. Stevens et al. formulated and evaluated folate receptor-targeted solid lipid nanoparticles of hematoporphyrin. It was found that targeted solid lipid nanoparticles gave IC50 of 1.57 micromolar whereas non-targeted solid lipid nanoparticles gave IC50 of 5.17 micromolar [74].

#### 3.2.3. Polymeric Micelles

Polymeric micelles have attracted attention in photodynamic therapy. They enhance aqueous solubility and retain photo–sensitizer in the blood for a longer period, allowing its accumulation at the desired site –enhanced permeability and retention. However, these photosensitizers loaded polymeric micelles can cause skin photosensitivity, damaging the blood vessel cells or the endothelial cells [127]. Lamch et al. fabricated folic acid conjugated polymeric micelles containing phthalocyanine for anticancer photodynamic therapy. The cell viability of free zinc phthalocyanine and FA conjugated zinc phthalocyanine polymeric micelles was found to be 65% and 59%, respectively in HaCaT cells. It was concluded that the synthesized functionalized polymeric micelles showed effectiveness in photodynamic therapy to treat metastatic melanoma and ovarian cancer [128]. Liu et al. developed EGa1 modified polymeric micelles for photodynamic therapy containing the photosensitizer meta-Tetra(hydroxyphenyl)chlorin. It was noted that the relative viability of photosensitizer-loaded polymeric micelles was found to be 25%, while that of EGa1-modified polymeric micelles showed relative viability of 10%. It was concluded that the developed nanoparticles enter the tumor cells upon binding of nanobody to EGFR receptors and result in enhancement of efficacy and selectivity of photodynamic therapy [129]. Lu et al. synthesized RGD-modified disulfide bond-conjugated prodrug polymer comprising of polyethylene glycol and camptothecin. The polymer self-assembled to form polymeric micelles. These micelles showed the potential to cross the blood-brain barrier and target the glioma cells. The normalized cell viability of CPD@IR780 was found to be 25% while that of only CPT was found to be 60%. It was concluded that photodynamic therapy and chemotherapy increased the efficacy of treatment [130].

## 4. Future Perspectives

PDT is entirely a non-invasive approach for treating cancer and this is its prime advantage over other conventional approaches such as surgery, curettage, cryosurgery etc., which are all invasive in nature. Additionally, PDT shows little to no side effects when compared to radiotherapy. Another advantage is that tumor cells have almost zero resistance to PDT, given that development of resistance is the major drawback associated with chemotherapy. We acknowledge that PDT as a treatment approach has a few disadvantages associated with it, such as development of phototoxicity post the treatment sessions. However, this occurs due to non-specific accumulation of photosensitizers and this challenge is being addressed by utilizing active nanoparticles which aim towards circumventing this problem. This helps in achieving reduced phototoxicity. PDT has been widely investigated in head and neck cancer, breast cancer, skin cancer, lung cancer, brain cancer, etc. Clinically, it has been in use for more than a decade now. Several clinical trials are ongoing in the present scenario where the effectiveness of PDT with different PS is being investigated in the treatment of cancers. In one clinical trial, the maximum tolerable dose of HPPH-PDT is being investigated in pleural malignancy [131]. In another similar clinical trial, the efficiency and safety of PDT with hexaminolevulinate (HAL) are being investigated in cervical neoplasia [132]. Despite all the clinical trials and positive results, a few disadvantages of classical PDT are yet to be overcome. The low water solubility of photosensitizer molecules and their poor tumor selectivity are the fundamental problems associated with classical PDT. The approach of incorporating nanocarriers into PDT is an effective strategy that has successfully solved these shortcomings of classical PDT. Surface modification of nanocarriers with PS molecules using aptamers, peptides, ligands, etc., is one of the most logical strategies to overcome the problems associated with conventional PDT therapy [61]. Additionally, several types of organic and inorganic nanomaterials combined with PS molecules result in the enhancement of water solubility of hydrophobic PS molecules as well as create a specific target approach for the delivery of the PS molecules via active and passive mechanisms [133]. However, fabrication targeted PS drug delivery parameters need to be thoroughly examined, including uses of the targeted moiety, biocompatibility of surface functionalization, loading capacity, and use of the specific wavelength of light for activation of PS. Moreover, significantly growing uses of nanomaterials for PDT face several obstacles in the way of clinical application such as long-term toxicity, systemic toxicity, phototoxicity, and dose-dependent toxicity. The development of high wavelength photoactive molecules needs to be fabricated in parallel with the development of actively targeted nanocarriers. Using a diverse set of multifunctional lipids, natural polymers, aptamers, peptides, etc., is an effective approach to avoid long-term and dose-dependent toxicity. Additionally, the use of site-responsive such as pH and temperature, etc., responsive drug delivery systems can overcome the problem associated with systemic and dose-dependent toxicity [134,135]. Several studies have also shown the effectiveness of using PDT and photothermal therapy (PTT) together. In this review, we have compiled various actively targeted organic and inorganic nanocarriers with targeted moiety to provide the site-specific delivery of PS molecules.

## 5. Conclusions

PDT is a promising treatment strategy in cancer treatment because of its non-invasive, patient-compliant, and highly effective nature. This review tries to present all the information available about actively and passively targeted NPs that are used in PDT specifically in cancer, in a concise manner. Serval cancer therapeutics strategy have been mentioned in an organized manner to give the reader a sense of fulfillment in the subject of using targeted nanocarriers in photodynamic therapy. To increase the site specificity, various means are being investigated continuously. The incidence of active as well as passive targeting via polymeric nanoparticles, inorganic nanoparticles, lipidic nanoparticles, and other novel nanocarriers are presented in the literature. However, a few common disadvantages are still prevalent with the nano strategy that need to be overcome in the future. These include unpredictable drug release profiles, inconsistent tumor permeation, long-term toxicity, etc. Other than this, the main hindrance to PDT is government body approvals. Numerous studies with PDT are still in clinical trial phases. Although backed by the scientific community, various factors have slowed the growth of the usage of PDT. The active interest of pharmaceutical companies and government bodies is needed to make this treatment strategy easy to follow for cancer patients. In conclusion, it can be said that incorporating targeted nanocarriers in PDT for cancer treatment shows enormous potential and possibility toward a more reliable, patient-compliant treatment strategy that should be explored further.

## Figures and Tables

**Figure 1 pharmaceutics-15-00109-f001:**
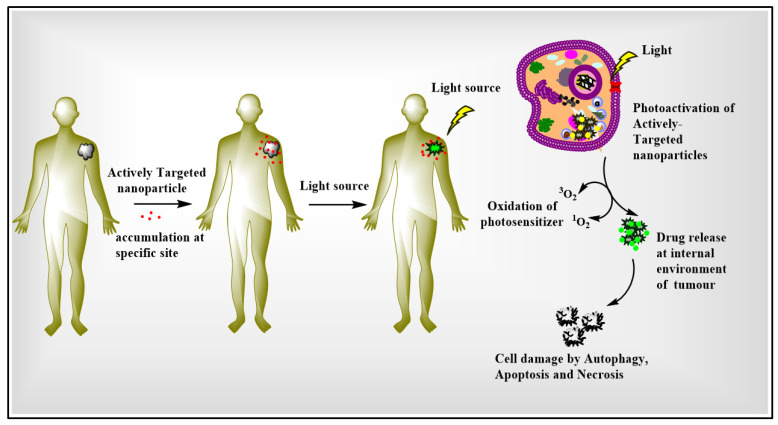
A general application of photodynamic therapy on solid tumors. Once the actively targeted nanoparticles enter the blood circulation, they are transported to the specific site of the tumor. The laser light irradiates the photosensitizer and activates the actively targeted nanoparticles in the tumor. Then, the photosensitizer molecules produce the singlet oxygen to kill the cancer cells by the mechanisms of apoptosis, autophagy, and necrosis, depending on the location of the photosensitizer in the tumor.

**Figure 2 pharmaceutics-15-00109-f002:**
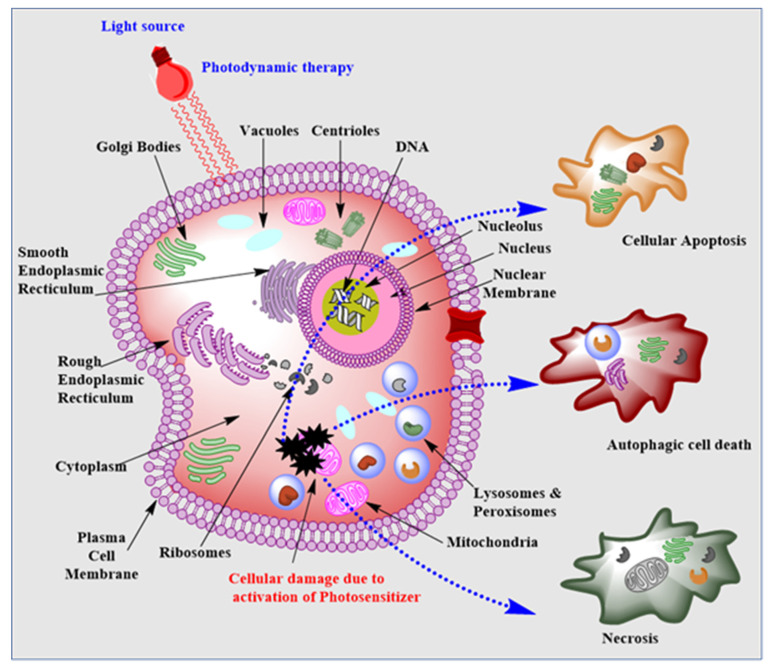
All the cellular organelles are depicted in this image. Upon irradiation, the PS molecules become activated and result in phototoxicity or death to the tumor tissue or cells through different mechanisms such as apoptosis, autophagy, and necrosis, depending on the internalization and localization of PS molecules into tumor tissues.

**Table 1 pharmaceutics-15-00109-t001:** Represents the clinical trials and approval status of various photosensitizers [30,31,32,33,34,35,36,37,38,39,40,41,42,43,44].

Sr. No	Photosensitizer with Generic Name	Chemical Structure	Activation Wavelength	Clinical Outcomes	References
1	Porfimer Sodium (Photofrin)	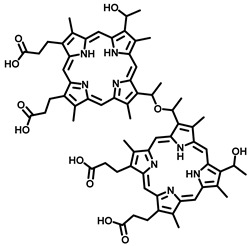	630 nm	Approved for bladder cancer, endobronchial cancer, esophageal cancer, lung cancer, and cervical cancer In clinical trial-brain cancer diagnosis	[30,31,37]
2	5-Aminolevulinic acid (Levulan and Ameluz)	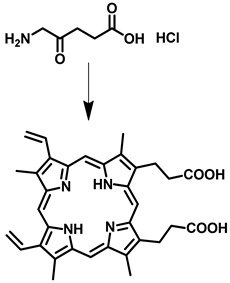	635 nm	Approved for non-melanoma skin cancers In clinical trial-brain cancer diagnosis	[38,39,40]
3	Hexaminolevulinate hydrochloride (Hexvix^®^)	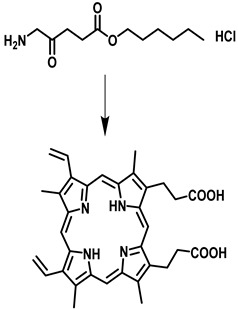	635 nm	Bladder cancer diagnosis	[41,42]
4	5,10,15,20-Tetrakis(3-hydroxyphenyl) chlorin/Temoporfin (Foscan)	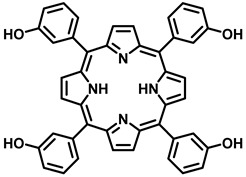	652 nm	Approved for head and neck cancer, prostate, and pancreatic cancer	[43,44]
5	2-(1-Hexyloxyethyl)-2-devinyl pyropheophorbide-a (HPPH) (PhotoChlor)	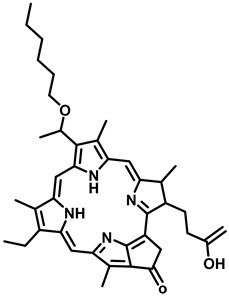	665 nm	In clinical trial-basal cell carcinoma, esophagus, mouth, and throat cancers, cervical intraepithelial neoplasia	[32,33]
6	Aluminum phthalocyanine tetrasulfonate chloride (Photosens)	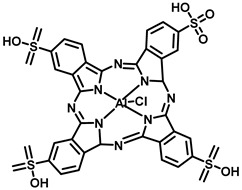	676 nm	Approved—Russia In clinical trials-stomach, skin, oral, and breast cancers	[34,35]
7	Palladium-Bacteriopheophorbde (WST09)/Padoporfin (Tookad)	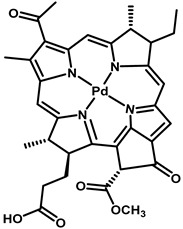	763 nm	Approved for prostate cancer in Mexico, Israel and 31 countries of the European Union	[36]

**Table 2 pharmaceutics-15-00109-t002:** Examples of recent major outcomes on nanoparticles-mediated photosensitizer delivery to the tumor by utilizing the overexpressed receptors site and their ligand.

Nanocarriers System	Photosensitizer	Tumor over Expression Receptor	Cancer Cell Lines/ Tumor Model	Major Outcomes	Ref
Methoxypoly(ethylene glycol)-poly(dl-lacticacid) (mPEG-b-PLA) Polymeric micelles	Chlorin e6	EGFR Receptor	A431 squamous carcinoma cell line & Xenografted mice model	Enhanced PDT effect and uptake with a targeting moiety, percentage of tumor growth inhibition (%TGI) was found to be (84%).	[76]
Poly(D,L-Lactide-co-glycolide (PLGA) & carboxymethyl chitosan (CMC) nanoparticle	Hypocrellin A (HA)	Transferrin-receptor	A549 human lung adenocarcinoma cell line & A549 tumor-bearing model Transferrin-receptor (TFR) positive	The tumor inhibition rate was found to be 63% for 15 days of Photodynamic therapy (PDT) treatment in tumor bearing mouse model.	[77]
Polymeric micelles	Meta-tetrahydroxyphenyl chlorin (m-THPC)	Folic acid receptor	KB oral cancer cell line & Xenografted animal model	m-THPC-loaded micelles were prepared by using a poly (2-ethyl-2-oxazoline)- b -poly(d, l -lactide) (PEOz-PLA) copolymer. A single dose of m-THPC with folic acid attached to polymeric micelles has shown antitumor effect against KB oral cancer cell line. % TGI rate was found to be 92% and an enhanced PDT effect was observed.	[78]
Porphyrin-lipid (porphysomes)	Porphyrin molecule	Folate receptor 1 (FOLR 1)	A549 & SBC5 lung cancer cells & mouse lung orthotopic tumor models	Porphyrin molecule lipid-based active targeted nanoparticle was prepared to enhance the efficacy and specificity of PDT by using FOLR 1. Folate-based porphysomes PDT significantly inhibited cell proliferation in lung cancer.	[79]
Gold Nanoparticles (Smart Cu(II)-aptamer complexes-based gold)	Chlorin e6	Specific targeting aptamers-TLS11a	Hepatocellular carcinoma cell line HepG2 and xenografted mouse model	These smart gold nanoparticles were fabricated for the programable synergistic targeted PDT therapy/Phototheramal therapy (PTT)/chemotherapy functionalities, to provide a synergistic effect for the treatment of hepatocellular carcinoma.	[80]
Titania Coated Upconversion NP Titanium dioxide (TiO2) poly ethylene glycol (PEG)up conversion	Chlorin e6	EGFR Receptor	CAL-27 oral squamous cell carcinoma (OSCC) & xenograft oral cancer tumor mouse model	Anti-EGFR-PEG-TiO2- upconversion nanoparticles as NIR PDT agents were fabricated for oral cancer. Enhanced intratumoral delivery and penetration to the deep thick tumor of head and neck cancer.	[81]
Polymeric NP	Chlorin e6 and Indocyanine green	Folic acid receptor and cyclic Arginyl-glycyl-aspartic acid (cRGD) targeting peptides	MCF-7 human breast cancer cell lines	The combination of PDT and Photothermal therapy has been proved to have the potential for tumor inhibition. Tumor apoptosis was found to be 85.9%.	[82]
Thermosensitive liposomes (TSL)	Chlorin e6	Folic acid receptor	Human cervical carcinoma (HeLa) cells & tumor mouse model	Thermosensitive dual drug-loaded nanocarrier for PDT with Photothermal therapy were prepared to utilize copper sulfide (CuS) and chlorin e6. Enhanced PDT effect and cellular uptake along with the controlled release of PS were observed. Higher phototoxicity and tumor inhibition was found.	[83]
Cationic dipeptide nanoparticle	Rose Bengal (RB) and bis(pyrene) (BP)	Cationic dipeptide (H-Phe-Phe-NH2_HCl,CDP)	MCF-7 breast cancer	(BP-CDPNP-RB) cationic dipeptide was fabricated for the one or two-photon-induced PDT therapy. Biocompatible and exhibit higher penetrability in tissue and enhances cellular uptake.	[84]
Chitosan/tripolyphosphate (TTP)	Curcumin	EGFR receptor	MKN45 human gastric cancer cells	Enhanced PDT effect, Curcumin nanoparticles were quickly taken up by the EGFR-overexpression cells.	[85]
Gold–Photoactive Polymer Nanoparticles (Gold acrylic copolymer with imidazole groups)	Spiropyran (SP)	Folic acid receptor	Rat brain C6 glioma cancer cell line	Photo responsive gold-decorated polymer nanoparticles (PGPNPs) were fabricated for improved PDT therapy and targeted efficiency towards rat brain cancer cell lines along with improved cellular uptake by 71.8%.	[86]
Polyacrylamide NP	Methylene blue (MB)	F3 peptides	GS-9L & F98 rat glioma & MDA-MB-435 human breast carcinoma cell lines	Tumor targeting MB-conjugated polyacrylamide nanoparticles were fabricated and used for active targeting, Results shows, that enhanced targeting effect and PDT efficacy with increasing dose and irradiation time.	[87]
Silica-based	Chlorin e6	Folic acid receptor	Human breast MDA-MB-231 cancer cells	Silica NPs were fabricated and coated with chlorin e6 and folic acid as a targeting agent. Results show that PDT and cellular uptake have been enhanced.	[88]

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
