# Peer review of "Multifunctional Photoactive Nanomaterials for Photodynamic Therapy against Tumor: Recent Advancements and Perspectives"

_pharmaceutics, 2022, doi:10.3390/pharmaceutics15010109_

Round 1

Reviewer 1 Report

The authors presents a Review paper about the delivery systems developed for the PTD accepted drugs and not only. The paper included a lot of new and well-known tumor specific molecules.

The only one which is in my research field is not included - Liposomes-delivery for photosensitizers such as naphthalocyanine (since 1993) and phthalocyanine complexes (papers Prof. G. Jori, 1992-2017). Zn(II)-phthalocyanine - DPPC liposomes was clinically accepted for PDT in Italy. Please include this effort!

Reviewer 2 Report

The manuscript entitled "Multifunctional Photoactive Nanomaterials for Photodynamic Therapy Against Tumor: Recent Advancements and Perspectives" is a diffuse and poorly organized treatment of the potential use of photodynamic therapy for the treatment of tumors. I have several major concerns that need to be addressed for the improvement of the manuscript.

1. The paper lacks a narrative and critical evaluation of the literature. Each single section of the manuscript is a list of the results of publications which lacks coherence, and a precise purpose.

2. There is considerable extraneous discussion unrelated to the topic. There is no clear indication exactly how tumors/diseases could be treated with nanoparticles in presence of PDT and how this might be successful.

3. It would have been better if the authors highlight the signaling cascades (cell death processes) and therapeutic targets in tumors by PDT. It would have given a better impression to the readers.

4. What are the added advantages of PDT as comparison with the conventional therapeutic strategies. What about toxicities associated with PDT?

5. The mitochondria-associated cell death modalities in presence of PDT would critically discuss by authors.

6. The informative graphical abstract would improve the manuscript.

7. In the manuscript, the following references may be considered:
DOI: 10.1016/j.pdpdt.2021.102697

DOI: 10.1016/j.bioactmat.2021.06.019

8. In the conclusion section, the authors need to discuss the key points and scope for further development. The research gap and open questions must be included.

9. Acronyms must be specified in the text, check through the entire paper to make sure it is defined at the first use.

10. It is better if the references replaced with the up to date references.

Reviewer 3 Report

Dear authors,

The paper submitted by Jain et al. approaches a very interesting topic and I hope that my remarks will help improving its’s quality.

1.     Fig.1 and Fig 2 – are original or require printing permission? Please include the cited sources in the description

2.     Please consider to include a table of abbreviations in the of the article. It will be easier for the reader to identify them.

3.     Please specify how did you select the articles and that your paper is a narrative review.

4.     Please adapt table 1 and 2 to the MDPI style and design.

5.     Line 179 – please add 3.1 in front of the name of the subchapter. Italic would be preffered for subchapters.

6.     Line 220 – please add 3.1.1 and so on further.

7.     Line 427 - please add 3.2 in front of the name of the subchapter.

8.     Line 428-– please add 3.2.1 and so on further.

9.     Line 499 – I would separate this section in two and keep a separate Conclusion section much more sythetic. The conclusions section usually has to include a „take home message” for the reader.

Overall, the manuscript is well written and organized. editing.

Best regards!

Reviewer 4 Report

The review: ” Multifunctional photoactive nanomaterials for photodynamic therapy against tumor: Recent advancements and perspectives” proposes a wide ranging report on the recent advancements about the use of photoactive nanomaterials for targeted tumor photodynamic therapy. The review is well written and only minor issues are to be checked, as detailed below, making the review to require a minor revision.

Lines 55-56 and Figure 1. Nanoparticles are here anticipated, in the middle of a general presentation of PDT. Therefore this should be moved down, where speaking of nanoparticles. On the contrary, and preferably, the figure could be maintained where it is, but correcting the “actively targeted” nanoparticle with a more general “photosensitizer”, and the related sentence changed accordingly,as for example: “ The application of  PDT on a solid tumor is schematically represented..”

Table 2 - The title of the column “Targeting moiety” is to be better explained in the test, or removed, since it can be already included in the next column. Only, at the convenience of the reader, an additional explanation  is to be provided for “mAB-cetuximab”, and for “mAB” in general

Round 2

Reviewer 2 Report

The authors addressed my concerns.